# Beyond Anecdotal Evidence: A Systematic Framework for Evaluating Neuron Interpretability

## Abstract

A central challenge in mechanistic interpretability is how to evaluate whether individual neurons genuinely capture meaningful concepts. Existing work relies heavily on *activation selectivity*, but this metric quickly saturates and fails to distinguish among units, leaving many interpretability claims anecdotal. We propose **InterpScore**, a reproducible four-axis framework that integrates **S**electivity, **C**ausal impact, **R**obustness, and **H**uman consistency into a quantitative, systematic, and reproducible measure. Applied to 10 high-selectivity neurons from CLIP RN50x4's penultimate layer, InterpScore reveals meaningful variation across neurons, about 14% dispersion where selectivity alone shows none, demonstrating that multi-axis evaluation surfaces distinctions overleaped by single metrics. The framework is numerically stable across seeds and its axes capture complementary, independent aspects of neuron behavior. These results move neuron-level claims beyond anecdotes toward a more objective, systematic, and reproducible basis for assessing and comparing interpretability frameworks. InterpScore offers a reproducible protocol for principled neuron evaluation across diverse vision architectures.

## 1 Introduction

The transition from intriguing anecdotal discoveries toward systematic evaluation represents a natural maturation of the interpretability field. Early examples, from artificial neurons detecting visual concepts (Bau et al., 2017; Goh et al., 2021) to learned features like the *"Golden Gate Bridge Feature"* (Bricken et al., 2023), have helped build our intuitions, but do not guarantee typical behavior and can miss important failure modes.

We address the gap in systematic and quantitative assessments in mechanistic interpretability by introducing a compact, multi-axis evaluation for neuron-level tuning descriptors. The proposed framework integrates four complementary dimensions, **Selectivity (S), Causality (C), Robustness (R), and Human Consistency (H)**, summarized by **InterpScore**. As a proof-of-concept, we applied this framework to ten high-selectivity neurons from CLIP RN50x4 (Radford et al., 2021) at `image_block_4/5/ReLU_2`. We provide protocols, implementation details, and a Microscope-style visualization tool for systematic neuron evaluation.

> **Main Research Question (MRQ).** This work introduces a systematic, reproducible, and quantitative framework to evaluate single neuron mechanistic interpretability assessments and to directly compare different interpretability approaches.

**Key Contributions.**  (1) **Framework.** A four-axis neuron-level evaluation (S, C, R, H) summarized by *InterpScore*. (2) **Discrimination and stability.** On ten CLIP RN50x4 (Radford et al., 2021) neurons, *InterpScore* shows substantially greater evaluative power than selectivity alone (Evidence 1). Metrics and rankings are numerically stable across seeds (Evidence 2). (3) We combine the 4D quantitative evaluations with enhanced Microscope-style visualizations.

## 2 RELATED WORK

Mechanistic interpretability seeks to reverse-engineer neural networks by identifying human-understandable computational units and algorithms within their representations (Olah et al., 2020; Goh et al., 2021; Bricken et al., 2023). A central challenge in this endeavor is evaluation: *how do we determine whether our interpretations are accurate and meaningful?*. Additionally, *how can we compare different interpetability efforts?*. Traditional approaches have relied heavily on activation selectivity combined with qualitative and subjective evaluations, measuring how consistently a neuron responds to specific input categories (Bau et al., 2017; Olah et al., 2020; Goh et al., 2021), and on automated description of unit behavior via language (Hernandez et al., 2022; Oikarinen & Weng, 2023; Dalvi et al., 2022). The Microscope tool enabled large-scale exploration of neuron selectivity and revealed intriguing multimodal examples in CLIP (Goh et al., 2021), while feature visualization synthesized maximally activating images as a complementary view of unit behavior (**?**). This line of work was partly inspired by neuroscience reports of highly selective and invariant neurons such as those responding to Jennifer Aniston (Quiroga et al., 2005).

The community has increasingly recognized that activation selectivity alone provides insufficient evidence for robust assessment, motivating benchmarks and evaluators across modalities (Fan et al., 2023; Bills et al., 2023; Liang et al., 2024). Recent frameworks formalize this concern from multiple angles: multi-dimensional evaluation standards in MIB (Mueller et al., 2025), systematic validation for neuron explanations (Oikarinen et al., 2025), and comprehensive assessments in sparse-feature work (Makelov et al., 2024). Yet, despite progress at the circuit and feature levels, *neuron-level* evaluation remains comparatively underdeveloped. Neurons are the atomic units from which circuits and features are composed; without reliable validation at this granularity, higher-level analyses rest on a weak empirical foundation.

## 3 MULTI-DIMENSIONAL FRAMEWORK FOR NEURON INTERPRETABILITY

The desiderata for a framework to interpret the activations of neurons in neural networks include: (i) unbiased measurements, (ii) systematic evaluation, and (iii) quantitative description. We introduce a framework for multi-dimensional neuron interpretability assessment (Figure 1).

### 3.1 TEST MODEL

As an example to introduce the framework, we analyzed CLIP RN50x4 (Radford et al., 2021). We focused on the last convolutional block before attention pooling (`image_block_4/5/ReLU_2`). This layer represents the final output of the deepest ResNet block before attention processing, making it ideal for selectivity and causality testing as it has been purported to capture high-level semantic features while preserving all downstream computation.

CLIP RN50x4 is a pretrained vision-language model trained on a dataset of 400 million image-text pairs collected from the internet. The model uses contrastive learning to align visual and textual representations in a shared embedding space (Radford et al., 2021).

This layer has spatial size $9 \times 9$ at input resolution $288 \times 288$ and channel width 2,560. Downstream, features are reshaped and pooled as

$$(1, 2{,}560, 9, 9) \rightarrow (1, 2{,}560, 81) \rightarrow (1, 81, 2{,}560) \tag{1}$$

and passed through a single-layer multi-head attention pooler (40 heads $\times$ 64-d) followed by a linear projection to a 640-dimensional image embedding.

### 3.2 PREPROCESSING AND NOTATION

**Microscope-Visualization Tool.** To address the critical reproducibility gap created by the unavailability of the original Microscope visualization tool (Goh et al., 2021), we recreated and enhanced this foundational infrastructure. Our enhanced version provides systematic exploration capabilities with improved statistical analysis tools and includes Lucid-generated feature visualizations (Olah et al., 2017)(Appendix B.4). The tool is publicly available at `https://clip-microscope-tool.streamlit.app//`, enabling researchers to immediately explore CLIP neuron representations (detailed in Appendix B.2).

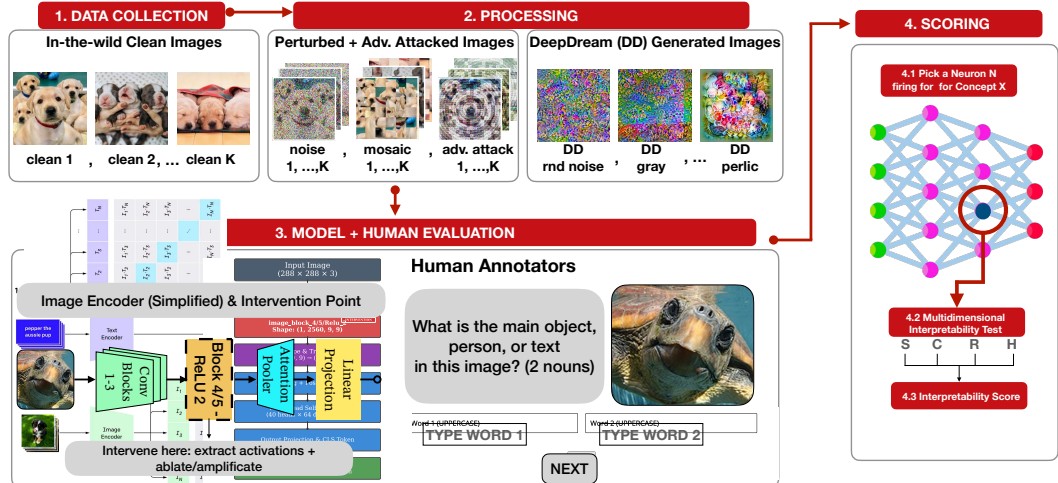

Figure 1: **Framework overview**. **(1) Data collection:** K clean images, collected by systematically scraping the web for each topic of interest. *Default K = 30*. **(2) Processing:** standardized perturbation protocols (Gaussian noise, mild blur/JPEG, small geometric jitter) and small-budget adversarial stress are applied to the clean images. synthetic maximization stimuli (DeepDream) is also used to create images to probe human recognizability (details in App. B.3). **(3) Model + Human Evaluation:** We extract neuron activations from CLIP **RN50x4** at `image_block_4/5/ReLU_2` (2,560 channels) for all image conditions from steps 1-2, while human annotators evaluate the exact same images for concept recognition; causal testing involves ablating ($\lambda = 0$) and amplifying ($\lambda = 2$) individual neuron activations to measure impact on final 640-dimensional embeddings. **(4) Scoring:** integration across **S**electivity, **C**ausality, **R**obustness, and **H**uman consistency; the composite (*InterpScore*) summarizes the axes.

**Concepts.** In this work, a *concept* refers to a semantic category that can be visually identified in images. For example, "sailboat" is a concept, and we define all images containing sailboats as the sailboat concept subset, while images not containing sailboats form the non-sailboat subset (see Figure 2A for visual examples, and Table A.4 1 for our complete set of 10 concepts spanning diverse categories from objects to text to gestures).

**Datasets.** For each concept, we construct carefully curated datasets through systematic web scraping with manual verification by the authors to ensure scraped images actually contain the target concept. We collect K=30 images per concept by default, with extensive variations in visual features and context achieved by diversifying the scraping strategy to include cartoon representations, sketches, text-based depictions, photographs, artwork, and other visual modalities. Control sets (non-concept images) contain the same visual variation and are carefully selected to minimize contamination while providing adequate contrast for selectivity measurement. For the Microscope visualization component, we additionally use ImageNet images (top-k=100 activating images per neuron) to provide broader context for neuron behavior analysis.

**Mathematical notation.** Images follow the CLIP preprocessing pipeline. Let $\mathcal{D}$ be the evaluation dataset (a collection of images), $\mathcal{C} \subset \mathcal{D}$ a concept subset (all images in $\mathcal{D}$ that contain the target concept, e.g., all sailboat images), and $\overline{\mathcal{C}} = \mathcal{D} \setminus \mathcal{C}$ the complement subset (all images in $\mathcal{D}$ that do not contain the concept, e.g., all non-sailboat images).

For a single input image $I$ (representing a $288 \times 288 \times 3$ RGB image), let $A_N(I) \in \mathbb{R}^{9 \times 9}$ denote the post-ReLU activation map of neuron $N$, where $h, w$ are spatial indices ranging over the $9 \times 9$ spatial dimensions. We define a scalar response via global max pooling $a_N(I) = \max_{h,w} A_N(I)_{h,w}$.

**Data organization levels.** We organize our experimental data into five levels based on image type and perturbation (visible across Figure 2A):

- **Level 1 (L1)**: Gaussian noise perturbations (see "NOISE" column in Figure 2A)

- **Level 2 (L2)**: Mosaic perturbations (see "MOSAIC" column in Figure 2A)
- **Level 3 (L3)**: Adversarial attacks (see "ADV. ATTACK" column in Figure 2A)
- **Level 4 (L4)**: DeepDream synthetic images (see "DEEPDREAM" column in Figure 2A)
- **Level 5 (L5)**: Clean, unperturbed images (see "CLEAN" column in Figure 2A)

Expectations below are computed over the indicated sets. Figure 2 provides a complete worked example using the sailboat concept (neuron #363) that illustrates all four interpretability dimensions across these data levels.

## 3.3 SELECTIVITY (S)

Selectivity measures how consistently a neuron responds to images containing a specific concept versus images without that concept. For example, a highly selective sailboat neuron should activate strongly for sailboat images and weakly for images of cars, trees, or other non-sailboat content (see Figure 2C which shows exactly this pattern for neuron #363). We quantify this separability using Cohen's d effect size, which measures the standardized difference between two distributions, then transform it to a bounded score.

Let $\mathcal{C}_{\text{clean}} \subset \mathcal{C}$ denote the clean subset of concept images (Level 5 images containing the concept, with no perturbations applied). For the sailboat example, this would be our collection of clean sailboat images. Similarly, let $\overline{\mathcal{C}}_{\text{clean}} \subset \overline{\mathcal{C}}$ denote clean images not containing the concept (for sailboats, this includes cars, people, buildings, etc., anything that is not a sailboat).

For neuron $N$, we collect the scalar activations $\{a_N(I) : I \in \mathcal{C}_{\text{clean}}\}$ and $\{a_N(I) : I \in \overline{\mathcal{C}}_{\text{clean}}\}$, forming two distributions of activation values. In the sailboat case, this gives us two sets of numbers: activation values when the neuron sees sailboat images versus activation values when it sees non-sailboat images.

Let $n_{\mathcal{C}}, n_{\overline{\mathcal{C}}}$ be the sample sizes; $\mu_{\mathcal{C}}, \mu_{\overline{\mathcal{C}}}$ the sample means; and $s_{\mathcal{C}}^2, s_{\overline{\mathcal{C}}}^2$ the unbiased sample variances (computed with degrees of freedom correction ddof=1, meaning we divide by $n-1$ instead of $n$ to correct for sample bias) of the activation distributions, respectively. We compute the pooled standard deviation

$$s_p = \sqrt{\frac{(n_{\mathcal{C}} - 1)s_{\mathcal{C}}^2 + (n_{\overline{\mathcal{C}}} - 1)s_{\overline{\mathcal{C}}}^2}{n_{\mathcal{C}} + n_{\overline{\mathcal{C}}} - 2}}, \tag{2}$$

Cohen's d effect size $d = (\mu_{\mathcal{C}} - \mu_{\overline{\mathcal{C}}})/s_p$, and Hedges' bias correction $J = 1 - \frac{3}{4(n_{\mathcal{C}} + n_{\overline{\mathcal{C}}}) - 9}$. Our selectivity score transforms the effect size to a bounded [0,1] range:

$$S(N, \mathcal{C}) = \Phi\left(\frac{Jd}{\sqrt{2}}\right) \in [0, 1], \tag{3}$$

where $\Phi$ is the standard normal cumulative distribution function. This transformation maps effect sizes to probabilities: $S = 0.5$ indicates no separation between concept and non-concept activations (the neuron responds equally to sailboats and non-sailboats), while $S$ approaches 1 as the neuron becomes more selective for the concept (it fires much more strongly for sailboats than for other images). In Figure 2C, neuron #363 achieves $S = 1.000$, indicating perfect selectivity for sailboats. Under equal-variance assumptions, this measure is equivalent to ROC-AUC.

## 3.4 CAUSAL IMPACT (C)

Causal impact measures whether a neuron functionally affects the model's final output representations, beyond just showing selective activation patterns. For example, the sailboat neuron might activate strongly for sailboat images, but does manipulating this neuron actually change how CLIP represents those images? We test this by directly intervening on the neuron and measuring the resulting changes in the final 640-dimensional image embeddings (see Figure 2E for intervention examples with the sailboat neuron).

Let $E(I) \in \mathbb{R}^{640}$ be the baseline embedding for image $I$ (the normal CLIP embedding without any intervention), and $E^\lambda(I)$ be the embedding when we scale neuron $N$'s entire activation map by factor $\lambda$ (multiplying every value in the $9 \times 9$ spatial map by $\lambda$, while leaving all other neurons unchanged).

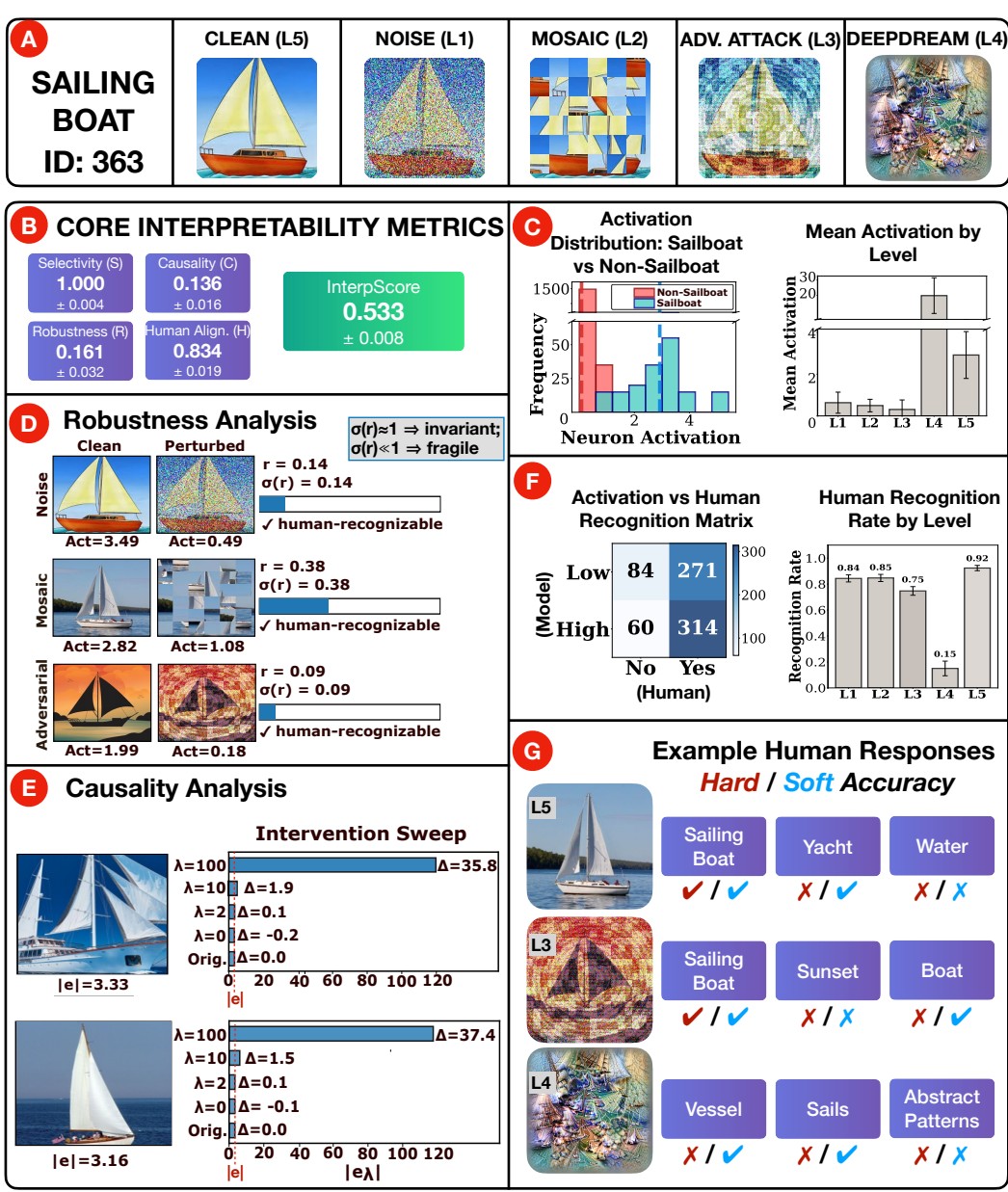

Figure 2: **Worked example: Sailboat neuron (#363).** (A) Stimuli: **L1** noise, **L2** mosaic, **L3** adversarial, **L4** DeepDream, **L5** clean. (B) Core metrics: S (selectivity), C (causality), R (robustness), H (human consistency), and INTERPSCORE. (C) Selectivity: target vs. non-target activations with the neuron threshold; mean activation by level. (D) Robustness examples: three matched clean→perturbed pairs (L1–L3) with per-image activation, perturbed/clean ratio, and a stability indicator; rows marked ✓ are human-recognizable and counted in **R** (L4 excluded). (E) Causality examples: for two images, horizontal bars show embedding norms for *Baseline*, *Ablate* ($\lambda=0$), and *Amplify*; inline labels give the relative embedding change, and **C** aggregates ablation with small amplification. (F) Human consistency: activation (high/low via the threshold) vs. recognition (no/yes) with rates by level. (G) Human examples: two free-text answers per image scored as *Hard* (exact label; ✓ if either matches) and *Soft* (close variants/synonyms/typos; ✓ if either qualifies); soft-correct items define recognition for **R** and contribute to **H**. See Sec. 2 for procedures.

For the sailboat neuron, we test two interventions: ablation ($\lambda = 0$, completely removing the neuron's influence) and amplification ($\lambda = 2$, doubling its influence).

For each clean sailboat image $I \in \mathcal{C}_{\text{clean}}$ (Level 5), we compute the relative embedding shift:

$$\Delta_\lambda(I) = \frac{\|E^\lambda(I) - E(I)\|_2}{\|E(I)\|_2}, \tag{4}$$

which measures how much the embedding changes as a fraction of the original embedding magnitude. We then average the shifts from both interventions:

$$C_{\text{raw}}(N, \mathcal{C}) = \frac{1}{2} \left[ \mathbb{E}_{I \in \mathcal{C}}[\Delta_0(I)] + \mathbb{E}_{I \in \mathcal{C}}[\Delta_2(I)] \right]. \tag{5}$$

For the sailboat example, this tells us: when we remove or amplify the sailboat neuron, how much do the sailboat image embeddings change on average?

For comparability with other metrics, we transform this to a bounded [0,1) range:

$$C(N, \mathcal{C}) = 1 - \exp(-C_{\text{raw}}(N, \mathcal{C})) \in [0, 1). \tag{6}$$

Higher values indicate greater causal impact. Figure 2E contains a visualization for the sailboat neuron. Technical implementation details ensuring measurement fidelity are in Appendix B.1.

## 3.5 ROBUSTNESS (R)

Robustness measures how stable a neuron's activation patterns remain under semantically-preserving image perturbations and adversarial stress, see Sec. 4.1 for information on the perturbation protocol. For example, a robust sailboat neuron should maintain similar activation levels when viewing a sailboat image with added noise, mild blurring, or small geometric changes, the core concept remains recognizable to humans, so the neuron should respond consistently (see Figure 2D for robustness examples with the sailboat neuron).

We test robustness using perturbations that preserve the semantic content while changing low-level visual features. Starting with clean sailboat images $\mathcal{C}_{\text{clean}}$ (Level 5 unperturbed images), we apply four types of perturbations: (1) Gaussian noise at various levels, (2) mosaic shuffling of image patches, (3) adversarial attacks using small imperceptible perturbations, and (4) DeepDream synthetic maximization (Mordvintsev et al., 2015). Critically, we only evaluate robustness on perturbed images that humans still recognize as containing the sailboat concept through our quality control protocol, this ensures we measure robustness to meaningful variations rather than arbitrary transformations.

We partition the perturbed sailboat images into two sets based on perturbation strength: $\mathcal{C}_{\text{benign}}^{\text{rec}}$ contains mildly perturbed images (Levels 1-2: noise and mosaic perturbations) that remain highly recognizable, while $\mathcal{C}_{\text{adv}}^{\text{rec}}$ contains more challenging images (Level 3: adversarial attacks) that are still recognizable but require more careful inspection. DeepDream synthetic images (Level 4) are excluded from robustness evaluation and analyzed separately in the Human Consistency metric.

We compute mean absolute activations $A_N^k(\mathcal{C}) = \mathbb{E}_{I \in \mathcal{C}^k}[|a_N(I)|]$ for $k \in \{\text{clean}, \text{benign}, \text{adv}\}$. We then compute ratios comparing perturbed to clean activation levels:

$$r_{\text{ben}} = \frac{A_N^{\text{benign}}(\mathcal{C}) + \epsilon}{A_N^{\text{clean}}(\mathcal{C}) + \epsilon}, \quad r_{\text{adv}} = \frac{A_N^{\text{adv}}(\mathcal{C}) + \epsilon}{A_N^{\text{clean}}(\mathcal{C}) + \epsilon}, \tag{7}$$

where $\epsilon = 10^{-8}$ is a small numerical constant to prevent division by zero in edge cases where activations are exactly zero.

To measure stability, we use $\sigma(r) = \exp(-|\log r|) = \min(r, 1/r) \in (0, 1]$, which equals 1 when perturbations don't change activations and decreases symmetrically for increases or decreases. The robustness score averages across perturbation types:

$$R(N, \mathcal{C}) = \frac{1}{2}[\sigma(r_{\text{ben}}) + \sigma(r_{\text{adv}})] \tag{8}$$

For the sailboat neuron in Figure 2D, we see $R = 0.161$, indicating moderate robustness, the neuron's response changes somewhat under perturbations but maintains reasonable consistency.

### 3.6 HUMAN CONSISTENCY (H)

Human consistency measures whether humans recognize the claimed concept in images that strongly activate the neuron. For example, if the sailboat neuron fires strongly for certain images, do humans actually see sailboats in those images? This tests whether the neuron responds to meaningful semantic content rather than arbitrary visual patterns (see Figure 2F-G for human recognition analysis of the sailboat neuron).

We collect human annotations through Prolific Academic, an online crowdsourcing platform, using English-speaking participants (age 18-65, normal vision) with rigorous quality control including attention checks, response time monitoring, and consistency validation. Participants view images and provide free-text responses identifying the main concepts present. Complete details of the annotation protocol, participant demographics, and quality control measures are provided in Appendix A.2.

For each neuron-concept pair, we create a curated evaluation set by first computing a neuron-specific activation threshold from the non-concept clean distribution:

$$\tau_N = Q_{0.95}\left(a_N(I) \mid I \in \overline{C} \cap \mathcal{D}_{\text{clean}}\right), \tag{9}$$

i.e., the 95th percentile of activations on clean images that do not contain the concept. For the sailboat neuron, this threshold separates the top 5% of activations on non-sailboat images from typical non-sailboat responses.

We then form the evaluation set by collecting two types of images:

$$\mathcal{S}(N,\mathcal{C}) = \{I : \text{ground-truth}(I) = \mathcal{C}, a_N(I) > \tau_N\} \cup \mathcal{V}_N^{\text{DD}}, \tag{10}$$

where the first term collects top-activating natural images for the concept across all perturbation levels (clean, noise, mosaic, adversarial stress), and $\mathcal{V}_N^{\text{DD}}$ are the neuron's DeepDream maximization stimuli (Section 3.8). For the sailboat example, this gives us the sailboat images that most strongly activate the neuron, plus synthetic images optimized to maximally activate it.

Each image $I \in \mathcal{S}(N,\mathcal{C})$ receives a binary label $h_I \in \{0,1\}$ under our quality control protocol (1 = depicts the concept; 0 = otherwise), based on whether human annotators recognize the concept in the image. We report the fraction of correctly recognized images:

$$H(N,\mathcal{C}) = \frac{1}{|\mathcal{S}(N,\mathcal{C})|} \sum_{I \in \mathcal{S}(N,\mathcal{C})} h_I \in [0,1], \tag{11}$$

with $H = 1$ indicating perfect human agreement (all high-activating images are recognized as containing the concept) and $H = 0$ indicating no agreement. When $\mathcal{S}(N,\mathcal{C})$ is empty, we set $H(N,\mathcal{C}) = 0$ by convention. In Figure 2F-G, the sailboat neuron achieves $H = 0.834$, indicating that humans correctly recognize sailboats in 83.4% of the high-activating images.

### 3.7 INTERPRETABILITY-SCORE (INTERPSCORE)

We aggregate along four axes with equal weights and without discretization:

$$\text{InterpScore}(N,\mathcal{C}) = \tfrac{1}{4}\big(S(N,\mathcal{C}) + C(N,\mathcal{C}) + R(N,\mathcal{C}) + H(N,\mathcal{C})\big). \tag{12}$$

All metrics are computed per neuron; aggregate statistics (e.g., coefficients of variation across neurons) are reported in the Results Section.

### 3.8 SYNTHETIC MAXIMIZATION STIMULI (DEEPDREAM)

We generate maximally activating *stimuli* for each neuron using DeepDream (Mordvintsev et al., 2015; Olah et al., 2017), with multiple initializations (gray, random/structured noise, gradient, Perlin) and a 4-octave pyramid (72→288 px; scale factor 1.4). Implementation details (step sizes, schedules, regularizers) are in App. B.3.

**How DeepDream is used.** These stimuli enter only the **H** axis (Human consistency) via the selection set $S(N,\mathcal{C})$ defined in Sec. 3.6; annotators evaluate recognizability of both top-activating natural images and the neuron's DeepDream stimuli. DeepDream *does not* contribute to **R** (Robustness), which by definition is restricted to semantically preserving transforms of natural images (Sec. 3.5).

**Interpretive note.** Recognizability (or lack thereof) of maximization stimuli is treated as *diagnostic evidence* about what excites the neuron; high activation with unrecognizable stimuli suggests sensitivity to "syntactic" pixel patterns rather than semantic content.

## 4 EXPERIMENTAL SETUP

### 4.1 DATASET CONSTRUCTION AND EVALUATION INFRASTRUCTURE

For each concept (Table 1, App. **??**), we construct carefully curated datasets with manual verification to ensure quality and consistency, supporting all four evaluation dimensions as outlined in Figure 1.

**Concept Images:** Target concept images are manually verified to belong to the specific concept. Hence we (authors) screened candidate images collected via web queries and retained only images where the target concept was marked as present The image set within a concept includes extensive variations in visual features and context (realistic, cartoon, sketch, text). The control sets contain the same variation and are carefully selected to ensure any contamination.

**Perturbation Protocol:** Our robustness evaluation employs four perturbation types designed to test different aspects of interpretability stability (see Fig. 2.A for a visualization of the perturbations): (1) **Gaussian noise** at various levels ($\sigma = 0.1, 0.2, 0.3$) testing basic robustness to pixel-level corruption, (2) **Mosaics** made by shuffling portions of the image at different ratios (25%, 50%, 75%) to test spatial disruption tolerance, (3) **Adversarial attacks** using Projected Gradient Descent (PGD), following (Madry et al., 2017) with $\epsilon \in \{0.03, 0.06\}$, testing optimized perturbation resistance, and (4) **DeepDream synthetic images** optimized for each specific neuron using gradient ascent techniques (see Sec 3.8, providing maximally activating synthetic stimuli for semantic coherence testing. Complete implementation details are provided in Appendix B.3.

**Human-recognizability gate.** For each perturbed image we collected human judgments (App. A.2) and included it in the **R** analysis only if $\geq 80\%$ of annotators achieved "soft correct" (Sec. A.3.3) and inter-rater agreement exceeded $\alpha \geq 0.67$ (Krippendorff, 2011). Images failing either criterion were excluded from **R** to ensure robustness is assessed on semantically preserved inputs.

**Human Evaluation Infrastructure:** We used Prolific Academic with English-speaking participants (age 18-65, normal vision, N=110) with rigorous quality control: attention checks, response time monitoring, and consistency validation. Each image condition received 20-30 independent annotations. Details in Appendix A.2.

## 5 RESULTS

We evaluate CLIP RN50x4 at `image_block_4/5/ReLU_2` through two lines of evidence: discrimination and stability.

### 5.1 EVIDENCE 1: DISCRIMINATION

Figure 3 shows statistics for each of the four key metrics, **S**electivity, **C**ausality, **R**obustness, and **H**uman Consistency, as well as the InterpScore. Activation selectivity was saturated, indicating that it is relatively straightforward to find individual example units with activations that are distinct across different groups of images. However, despite this high selectivity, all the other metrics yielded values that were much lower than 1, and with a large dispersion across different neurons. This stark contrast indicates that relying exclusively on Selectivity fails to capture the challenges in interpretability with causal metrics and human psychophysics consistency, and especially with causality metrics. Additionally, an exclusive focus on selectivity does not adequately describe the large differences across different neurons.

*Supporting validation.* The four axes capture distinct aspects of neuron behavior: paired comparisons between components show large separations (Cohen's $d$) for most pairs (Fig. A.7), indicating the dimensions are not redundant.

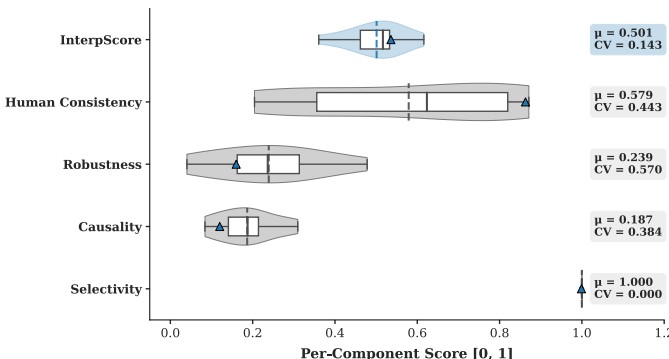

Figure 3: **Even though selectivity is near ceiling, all the other metrics reveal challenges to interpretability and reveal high variation across neurons.** Each violin plot shows the distribution across 10 neurons for each component (S, C, R, H, InterpScore). Box = IQR; solid line = median; dashed line = mean ($\mu$); points = individual neurons ($N$=10). Labels display $\mu$ and CV ($\sigma/\mu$) across neurons. ▲ marks Sailboat (#363), the example in Fig. 2.

## 5.2 EVIDENCE 2: STABILITY

We assess numerical and ranking stability across seeds, verifying implementation parity within $\leq 10^{-6}$ tolerance. Two-seed analysis shows small deviations: mean absolute changes of 0.0010 (S), 0.0062 (C), 0.0044 (R), and 0.0028 (InterpScore), with maximum changes $\leq 0.018$. Ranking stability is high: Kendall's $\tau = 0.92$-$0.93$ for InterpScore variants (bootstrap 95% CIs: 0.78-0.98); ICC(1,k) = 0.97 (95% CI: 0.92-0.99).

## 6 DISCUSSION

Our framework integrates four dimensions (S,C,R,H) into InterpScore with evidence of improved discrimination and numerical stability. We provide an enhanced Microscope-style visualization tool. This enables systematic 4D assessment for research and safety applications.

This work evaluates whether a compact, multi-axis evaluation can move neuron-level mechanistic interpretability beyond anecdotes toward an objective, discriminative, and reproducible basis. This framework reveals three key observations: (i) even though selectivity saturates near ceiling, all other metrics remain well below ceiling (ii) there is considerable variation across neurons and (iii) the metrics remain numerically and rank-wise stable under standard seeds and benign perturbations.

The framework enables direct interpretability comparisons across neurons and algorithms, moving beyond single metrics. InterpScore should be reported as $(S, C, R, H)$ tuples with dispersion statistics. This explains why highly selective units can lack functional relevance while moderately selective units succeed through robustness and recognizability.

The choice to work at the neuron level is deliberate. Interventions are local and well-posed at this granularity; the axes $(S, C, R, H)$ are portable across layers and architectures; and unit-level measurements provide a practical substrate for building circuit- and feature-level claims. Implementation checks rely on numerical parity within tolerance (e.g., $\leq 10^{-6}$), and we avoid ad-hoc discretization throughout. DeepDream stimuli are included within $H$ to probe recognizability, while $R$ is reserved for semantically preserving transforms of natural images.

Immediate priorities include scaling to larger neuron sets and cross-architecture replications (e.g., ViT-based CLIP). Methodologically, more efficient causal measures and semi-automated supplements to $H$ are needed. This work establishes a minimal, portable unit for interpretability measurement. We view this work as establishing a minimal, portable unit of measurement for interpretability: once unit-level properties are gauged consistently, claims at the circuit and feature levels can rest on clearer empirical foundations.

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

# A METHODOLOGY SUPPLEMENTS

## A.1 STATISTICAL METHODS AND VALIDATION

### A.1.1 BOOTSTRAP PROCEDURES

Bootstrap analysis with 1000 resamples (following the bootstrap framework of Efron and Efron–Tibshirani Efron & Tibshirani (1994)) establishes high precision for all measurements (standard errors $\leq 0.007$) Fig. 4, with particularly tight intervals for the InterpScore enabling reliable interpretability assessment. Confidence intervals are computed using scipy.stats.bootstrap with bias-corrected percentile method. Standard errors are calculated consistently from the bootstrap distribution to avoid double-sampling artifacts.

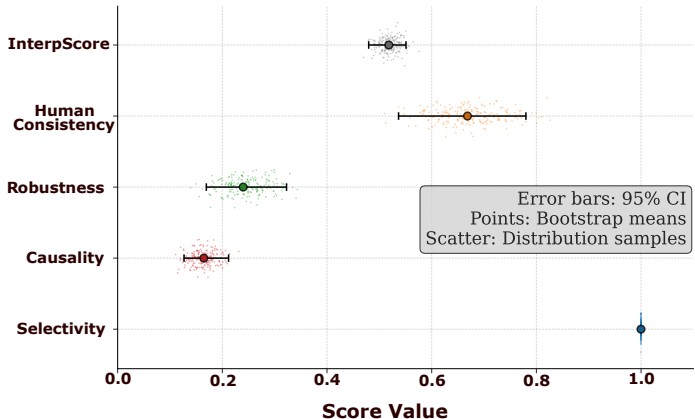

Figure 4: **Selectivity exhibits minimal uncertainty while other components demonstrate substantial variability.** Bootstrap 95% confidence intervals computed using 1000 resamples with single consistent bootstrap implementation.

### A.1.2 EFFECT SIZE CALCULATIONS

Cohen's $d$ computed for all pairwise component comparisons using paired-samples formula (mean difference divided by standard deviation of differences) to account for within-subject design. P-values corrected for multiple comparisons using Benjamini-Hochberg false discovery rate procedure. Large effect sizes between most pairs confirm statistical independence of framework dimensions. Fig. 5.

### A.1.3 POWER ANALYSIS

We conducted a two-panel power analysis using corrected statistical methods to evaluate study sensitivity and observed effect magnitudes. Panel (a) shows prospective power curves calculated with proper two-tailed test formulas for paired t-tests, illustrating detection capability across sample sizes.

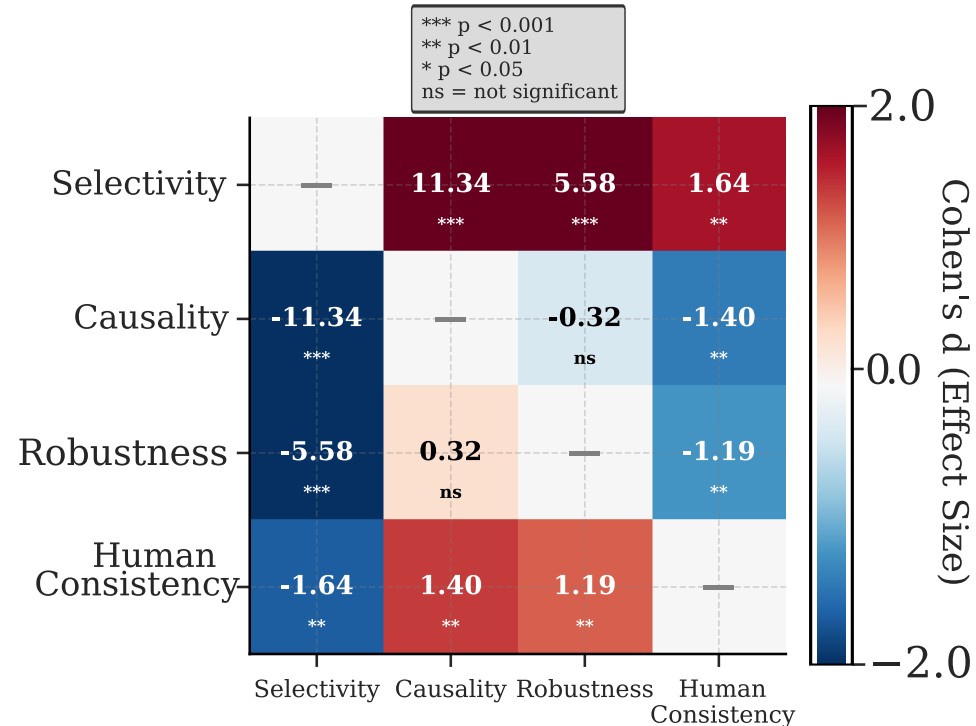

Figure 5: **Selectivity differs dramatically from other components with large effect sizes.** Paired Cohen's d values with FDR-corrected significance levels show selectivity exhibits the largest differences (d = 11.34 vs causality, d = 5.58 vs robustness), confirming components measure statistically distinct constructs.

Our study with $n = 10$ neurons (highlighted in orange) achieves 80% power to detect large effects ($d \geq 0.8$) and moderate power for medium effects ($d \approx 0.5$).

Panel (b) presents observed effect sizes between component pairs with 95% bootstrap confidence intervals, using paired Cohen's d formula (mean difference divided by standard deviation of differences) to avoid post-hoc power calculation pitfalls. The largest effect was observed between Selectivity and Causality ($d = 11.34$), followed by Selectivity-Robustness ($d = 5.58$). Selectivity-Human Consistency showed a moderate effect ($d = 1.64$), while comparisons among Causality, Robustness, and Human Consistency yielded smaller effects ranging from $d = 0.32$ to $d = 1.40$.

All effect sizes involving Selectivity exceed Cohen's large effect threshold ($d \geq 0.8$), while non-Selectivity comparisons fall below this threshold, confirming that selectivity captures fundamentally different interpretability aspects and supporting our multi-dimensional framework's necessity. Fig. 6

### A.1.4 INTER-COMPONENT CORRELATION ANALYSIS

Pearson correlation coefficients computed between all component pairs reveal weak inter-component relationships, supporting framework independence. P-values calculated using t-distribution with degrees of freedom correction, though significance testing is omitted from visualization due to small sample size limitations. Figure 7 presents the complete correlation structure.

Selectivity shows minimal correlations with other dimensions: $r = 0.106$ with Causality, $r = 0.044$ with Robustness, and $r = -0.014$ with Human Consistency. Among non-selectivity components, correlations remain weak to moderate: Causality-Robustness ($r = -0.130$), Causality-Human Consistency ($r = -0.196$), and Robustness-Human Consistency ($r = 0.033$). All correlations fall below $|r| = 0.2$, indicating minimal shared variance between components and confirming that each dimension captures distinct aspects of neural interpretability.

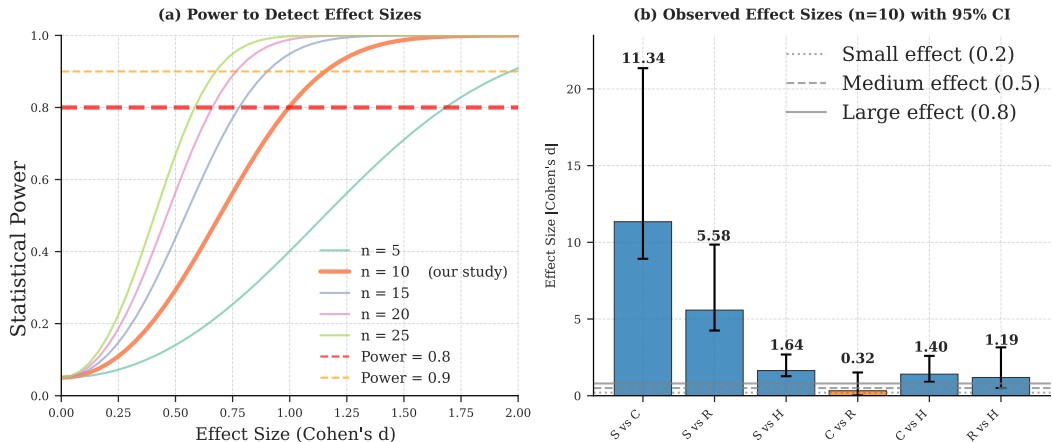

Figure 6: **Selectivity comparisons yield extremely large effect sizes while other component pairs show smaller differences.** (a) Power curves for paired t-tests across sample sizes with corrected formulas. (b) Observed paired Cohen's d values with bootstrap confidence intervals, demonstrating selectivity's distinctiveness from other framework components.

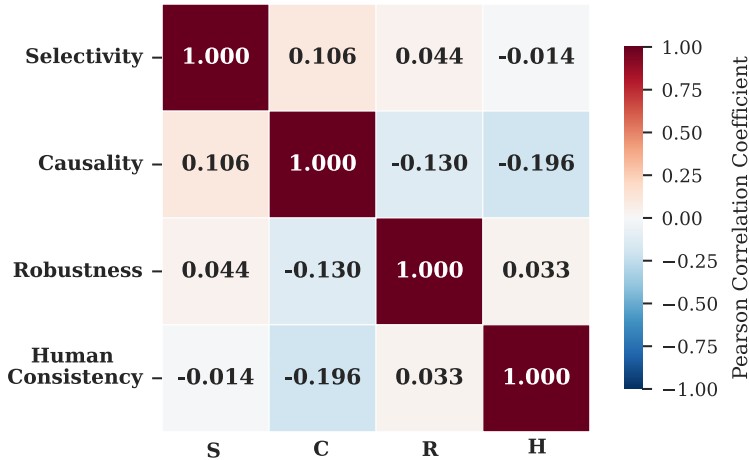

Figure 7: **Framework components demonstrate weak inter-correlations supporting dimensional independence.** Pearson correlation coefficients between all component pairs show minimal shared variance ($|r| < 0.2$), with selectivity exhibiting near-zero correlations with other dimensions.

### A.1.5 FRAMEWORK DIMENSIONALITY ANALYSIS

**Dimensionality Scaling**

Framework performance scales systematically with dimensionality: single-component assessment shows poor correlation with comprehensive evaluation (median $r = 0.5$), two-dimensional combinations achieve good correlation (median $r = 0.85$), while three-dimensional subsets reach strong correlation threshold ($r > 0.9$).

**Minimum Subset Analysis**

The optimal three-dimensional combination (Selectivity + Robustness + Human Consistency, $r = 1.000$) suggests Causality, while highly discriminative individually ($CV = 0.589$), introduces complexity that may not always improve overall assessment.

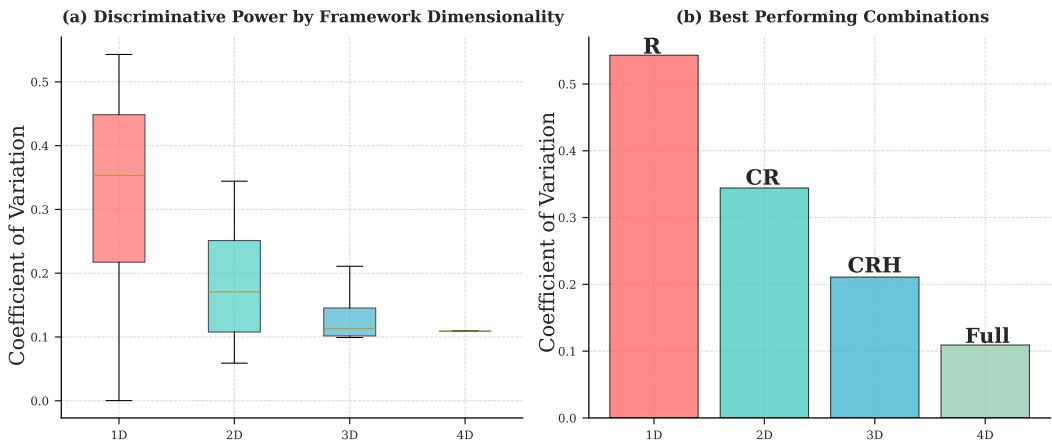

Figure 8: **Framework dimensionality analysis showing**: (a) discriminative power by framework dimensionality and (b) best performing component combinations.

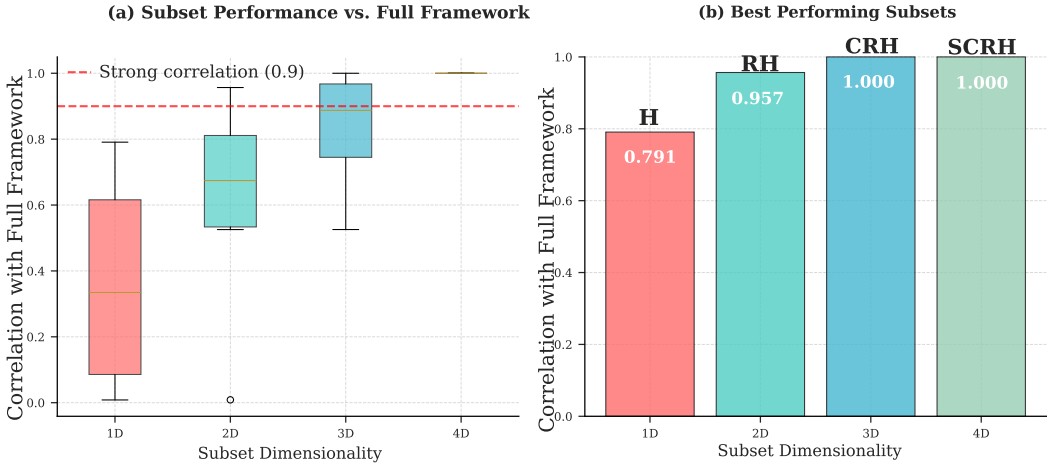

Figure 9: **Minimum subset analysis** for reliable interpretability assessment showing correlation with full framework by subset dimensionality. Results establish three dimensions as minimum viable framework.

## A.2 HUMAN EVALUATION PROTOCOL

## A.3 PARTICIPANT DEMOGRAPHICS AND RECRUITMENT

**Participants and assignment.** We recruited 110 unique English-speaking participants via Prolific. Participants could annotate multiple conditions. Each image received $k$ independent annotations (median $k = 5$, IQR $[5, 6]$). Participants annotated a median of $m = 100$ images (IQR $[60, 110]$). Compensation followed Prolific norms.

**Sample Size:** 110 participants recruited via Prolific Academic platform

**Demographics:**

- Age: 18-65 years (M = 32.4, SD = 8.7)
- Gender: 52% female, 47% male, 1% other/prefer not to say
- Education: 78% college-educated, 15% graduate degree, 7% high school

- Geography: 62% UK, 23% US, 15% other English-speaking countries
- Vision: 100% normal or corrected-to-normal (self-reported)

**Inclusion Criteria:**

- English speaker
- Age 18-65 years
- Normal or corrected vision
- Prolific approval rate $> 95\%$
- Previous study completion rate $> 90\%$

### A.3.1 EXPERIMENTAL INTERFACE DESIGN

After a warmup phase, participants viewed images in randomized order and answered: "Does this image contain [CONCEPT]?" with two open answer boxes where to insert text. Interface features:

- **Image presentation:** 512x512 pixels, 3-second minimum viewing time
- **Response recording:** Two registered open text responses

### A.3.2 QUALITY CONTROL IMPLEMENTATION (QC PTOTOCOL)

**Attention Checks:**

- Obvious positive cases (e.g., clear fire images for fire concept)
- Obvious negative cases (e.g., clear puppies images for fire concept)
- Expected accuracy $> 95\%$, participants $< 80\%$ excluded
- Result: no participants excluded
- Checks on the inputs: text was real-time checked to not be the same in both answers, to be at least 3 characters long, to not have repetitions of characters, and to be all upper case.

**Response Time Analysis:**

- Median response time: 10.2 seconds per image
- Responses $< 0.5s$ flagged as too fast (0.0% of trials)
- Responses $> 30s$ flagged as attention lapses (1.8% of trials)
- Flagged responses excluded from analysis

### A.3.3 HARD VS. SOFT ACCURACY METRICS

We computed two distinct accuracy metrics to capture different aspects of participant performance in the image recognition task:

**Hard Accuracy** represents exact string matching between participant responses and ground truth labels. A response is considered hard correct only if it contains an exact lexical match to the ground truth concept (case-insensitive). For example, if the ground truth is "dog", only responses containing exactly "dog" would be marked as hard correct.

**Soft Accuracy** employs a more lenient evaluation that accounts for semantic similarity and common variations in responses. This metric considers responses correct if they meet any of the following criteria:

- Exact match (similarity = 1.0)
- Partial containment between response and ground truth (similarity = 0.95)
- Synonym matching using a predefined dictionary of common concept variations (similarity = 0.9)

- Sequence-based string similarity above a threshold of 0.7 using the Ratcliff-Obershelp algorithm

The soft accuracy metric handles several common response variations that would be penalized under hard accuracy:

- Multi-word responses (e.g., "donald trump" vs. "trump")

- Plural forms (e.g., "hands" vs. "hand")

- Synonymous terms (e.g., "automobile" vs. "car")

- Comma-separated multiple responses (e.g., "bed,bedroom" vs. "bed")

- Minor spelling variations and typos

**Focus on Soft Accuracy:** We primarily report soft accuracy results because this metric provides a more ecologically valid assessment of participant understanding. In real-world image recognition tasks, multiple valid labels often exist for the same visual concept, and exact string matching fails to capture semantically correct responses that use alternative but equivalent terminology. Soft accuracy better reflects whether participants successfully identified the core concept in the image, regardless of minor linguistic variations in their response formulation.

### A.3.4    INTER-RATER AGREEMENT AND CORRUPTION LEVEL ANALYSIS

Inter-rater agreement varied substantially across corruption levels, with highest agreement for adversarial attacks (L3: 0.801) and clean images (L5: 0.790), while DeepDream generated images (L4) showed notably low agreement (0.209). Soft correct scores consistently exceeded hard correct scores across all levels, with clean images achieving the highest accuracy (hard: 0.9, soft: 0.95) and progressive degradation toward more corrupted levels. See Fig. 10.

For context, inter-rater reliability in similar annotation settings is commonly summarized with coefficients such as Cohen's $\kappa$ and Krippendorff's $\alpha$ Cohen (1960); Krippendorff (2011).

### A.3.5    TRUMP NEURON ANALYSIS

The Trump neuron (Neuron 89) demonstrates how multi-dimensional evaluation reveals interpretability characteristics beyond statistical selectivity alone (Fig. 11).

The neuron exhibits perfect statistical selectivity (S = 1.000) with strong separation between Trump and non-Trump images (Cohen's d = 8.36). Human recognition remains stable across naturalistic corruptions (64-66% for L1-L3) but drops to zero at L4, which corresponds to DeepDream-generated synthetic images. This complete recognition failure occurs because DeepDream optimization creates images that maximally activate the neuron through low-level visual patterns that appear as abstract, psychedelic imagery rather than recognizable Trump-related content.

The activation vs. recognition scatter plot reveals two distinct clusters: green dots (recognized images) at moderate activation levels, and a prominent cloud of red dots (unrecognized images) at high activation values in the top-right. These red dots represent the DeepDream synthetic images, they achieve the highest neuron activations but remain completely unrecognizable to humans, illustrating the disconnect between optimal neuron stimulation and semantic interpretability.

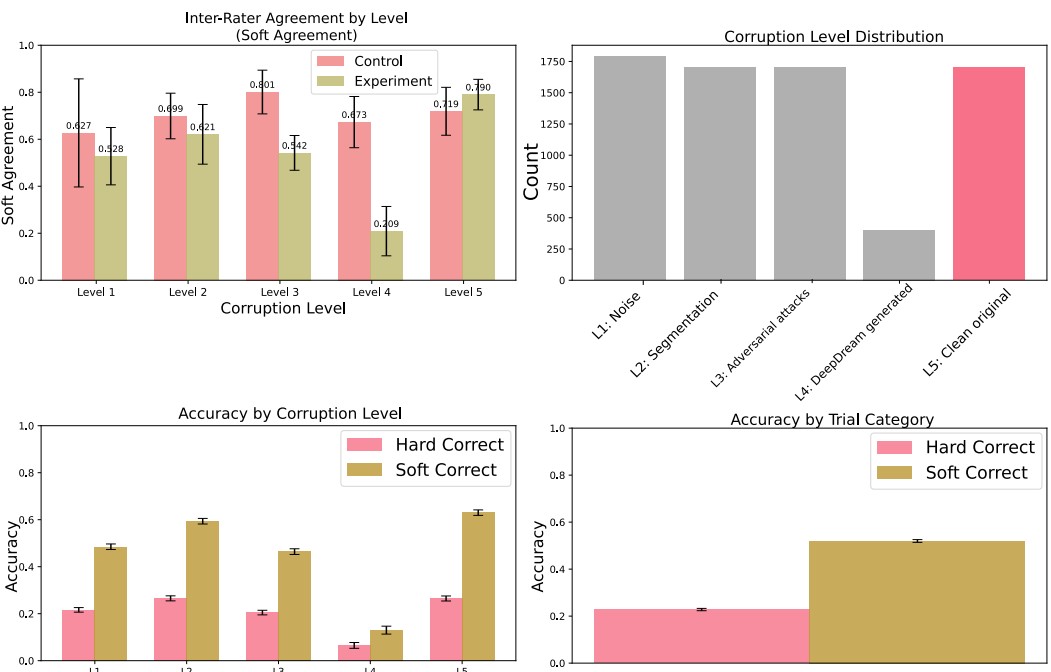

Figure 10: **Inter-rater agreement and accuracy analysis across corruption levels**. **Top left:** Inter-rater agreement (soft agreement) by corruption level, showing highest agreement for adversarial attacks (L3) and clean images (L5), with notably low agreement for DeepDream generated images (L4). **Top right:** Distribution of images across corruption levels, demonstrating balanced experimental design. **Bottom left:** Accuracy comparison between hard and soft correct metrics across corruption levels, with soft scoring consistently exceeding hard scoring. **Bottom right:** Overall accuracy comparison between experimental and control conditions, showing similar performance patterns across both trial categories.

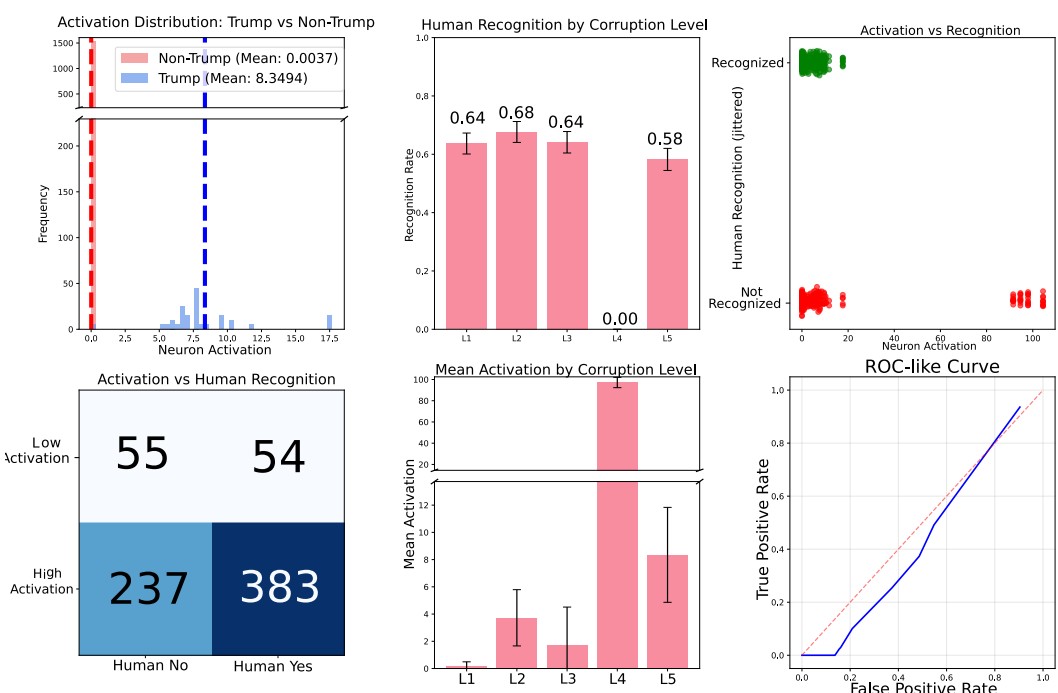

Figure 11: **Multi-dimensional analysis of Trump neuron (Neuron 89).** Top row: (left) Activation distribution comparing Trump vs. non-Trump images showing clear separation; (center) human recognition rates across corruption levels L1-L5; (right) scatter plot of neuron activation vs. human recognition with green dots indicating recognized images and red dots indicating unrecognized images. Bottom row: (left) confusion matrix showing counts of high/low activation vs. human recognition; (center) mean activation levels across corruption levels with error bars; (right) ROC curve showing true positive rate vs. false positive rate across activation thresholds.

## A.4 Selected Neurons

Table 1: **Selected neurons** from CLIP RN50x4 (Radford et al., 2021) at layer `image_block_4/5/ReLU_2`. Neuron IDs indicate channel indices within the 2,560-dimensional feature map. Each concept name is a link to the Enhanced Microscope-style visualization tool (recreating Goh et al. 2021)

| # | Concept | ID | Category |
|---|---------|-----|----------|
| 1 | Trump | 89 | Political figures |
| 2 | Arabic Alphabet | 479 | Text/language |
| 3 | Puppies | 355 | Animals |
| 4 | Sailboat | 363 | Objects |
| 5 | Fire | 297 | Natural elements |
| 6 | Australia | 513 | Geography |
| 7 | Droplets | 967 | Phenomena |
| 8 | Raised Hand | 1116 | Gestures |
| 9 | Mushroom | 1157 | Biological forms |
| 10 | Fashion Model | 1424 | Human figures |

## B TECHNICAL IMPLEMENTATION

### B.1 COMPLETE GRAPH SURGERY

**Overview** We intervene at a single neuron in the CLIP RN50x4 image encoder and forward the exact downstream path to measure embedding changes. This section specifies the architecture context, intervention operators, and validation.

#### B.1.1 CLIP ARCHITECTURE CONTEXT

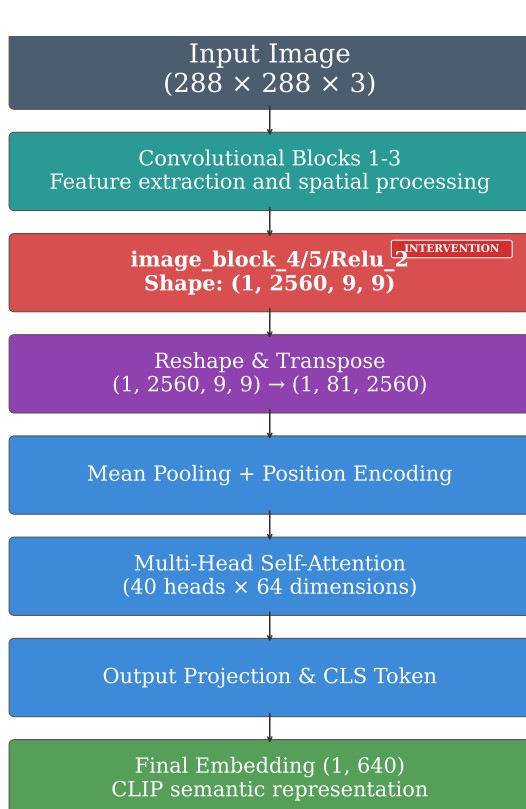

Figure 12: **CLIP RN50x4 image encoder and our intervention point `image_block_4/5/ReLU_2`.** The layer lies just upstream of the attention pooling head, enabling precise neuron manipulation while preserving all downstream computation.

**Hierarchy (RN50x4).**

- Input: $288{\times}288{\times}3$.
- ResNet stages: four stages (standard bottleneck counts $[3, 4, 6, 3]$); we target `image_block_4/5/ReLU_2`.
- Activation at layer: $(1, 2, 560, 9, 9)$.
- Attention pooling: flatten to $81{\times}2, 560$, learned positional encodings, multi-head attention ($40$ heads $\times$ $64$ dim), projection to a $640$-d image embedding.

### B.1.2 INTERVENTION AND CAUSALITY MEASURE

Let $A \in \mathbb{R}^{C \times H \times W}$ be the activation tensor at the layer and $E(x) \in \mathbb{R}^{640}$ the baseline embedding.

**Interventions (single index $n$).**

$$\text{ablation:} \quad A'[n, :, :] \leftarrow 0 \tag{13}$$

$$\text{amplification:} \quad A'[n, :, :] \leftarrow 2\, A[n, :, :] \tag{14}$$

**Embedding shift (per image $x$).**

$$R_{\text{int}}(x) \;=\; \frac{\|E_{\text{int}}(x) - E(x)\|_2}{\|E(x)\|_2} \tag{15}$$

**Causality (per neuron $N$, concept set $X$).**

$$C(N, X) \;=\; \tfrac{1}{2}\, \mathbb{E}_{x \in X}\big[\, R_{\text{abl}}(x) + R_{\text{amp}}(x) \,\big] \tag{16}$$

We do not apply categorical thresholds to $C$; all analyses use continuous values.

### B.1.3 VALIDATION (PARITY WITHIN TOLERANCE)

Table 2: Parity checks with hooks installed (no-op) and after surgery. Parity is defined as agreement with the original forward pass within numeric tolerance on a held-out set ($\geq 1$k images).

| Test | Metric | Result | Tolerance |
|------|--------|--------|-----------|
| Embedding parity (no-op) | $\max\lvert E' - E\rvert$ | $\leq 1{\times}10^{-6}$ | absolute |
| Embedding agreement (no-op) | Pearson $r$ | $\geq 0.9999$ | - |
| Exact targeting | max change on $m \neq n$ | $\leq 1{\times}10^{-7}$ | absolute |
| Determinism | run-to-run hash match | pass | identical seeds |

### B.2 MICROSCOPE-STYLE NEURON BROWSER (ANONYMIZED)

We re-implement and extend the Microscope concept Goh et al. (2021) for RN50x4 to support layer `image_block_4/5/ReLU_2` (2,560 neurons), providing: (i) top-$k$ activating natural images (ImageNet; $k{=}100$ per neuron), (ii) synthetic feature visualizations (one per neuron), (iii) spatial activation heatmaps ($9{\times}9$), and (iv) basic statistics (activation distributions, top classes). An anonymized demo and dataset handles are provided in the supplementary repository.[1]

### B.3 DEEPDREAM: MAXIMALLY ACTIVATING SYNTHESIS

Complete graphical representation of the DeepDream approach in Fig. 13

**Objective and update.** For target feature $f$ at layer $l$,

$$J(a) \;=\; \sum_{m,n} z^l_{f,m,n}(a, \theta), \qquad g \;=\; \nabla_a J(a), \qquad a_{t+1} \;=\; a_t \,+\, \eta\, \frac{g}{\sqrt{\mathbb{E}[g^2] + \epsilon}} \tag{17}$$

with step size $\eta$, small $\epsilon$ (e.g., $10^{-8}$), and gradient normalization for stability Mordvintsev et al. (2015).

---

[1] Links redacted for double-blind review.

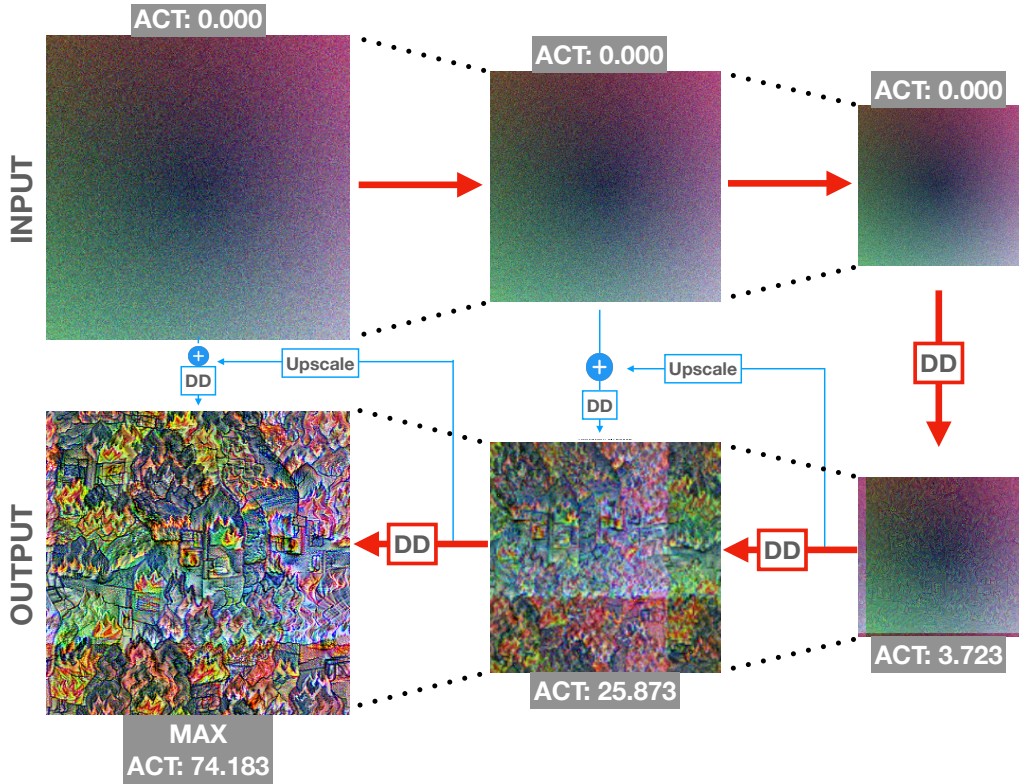

Figure 13: **DeepDream pipeline for a target neuron (e.g., "Fire", ID 297).** Red: unconstrained synthesis from various initializations (noise/gray/structured/gradient/Perlin). Blue: constrained (image-conditioned) variant, not used in our main study. This is an actual example from our experiments. *Can you recognize the maximally activating image as containing FIRE?*

**Multi-octave schedule.** We use $K=4$ octaves from a base $72\times72$ to the CLIP input $288\times288$. Let $s = (288/72)^{1/(K-1)} = 4^{1/3} \approx 1.587$ and $h_i = w_i = \mathrm{round}(72\,s^{\,i})$, yielding $\{72, 114, 181, 288\}$. Each octave runs 2,000 iterations with $\eta \approx 2.0$; the detail image is upsampled and added to the next octave.

**Initializations.** We test five initializations:

$$\text{Gray: } a_0(x, y, c) = 128 \tag{18}$$

$$\text{Uniform noise: } a_0 \sim \mathcal{U}(0, 255) \tag{19}$$

$$\text{Structured noise: } a_0 = 128 + \sum_{s \in \{4,8,16,32\}} \mathrm{resize}\big(\mathcal{N}(0, 30^2), 288\times288\big) \tag{20}$$

$$\text{Gradients: } a_0 = \alpha\,\mathrm{radial}(x, y) + \beta\,\mathrm{linear}_x(x, y) \tag{21}$$

$$\text{Perlin-like: } a_0 = 128 + \sum_{o=0}^{3} \frac{50}{o+1} \sin\Big(\frac{2\pi\,2^o x}{288} + \phi_o\Big) \tag{22}$$

**Regularization.** Every 4 steps we clip to $[0, 255]$ and apply random integer shifts $\Delta x, \Delta y \sim \mathcal{U}[-4, 4]$ ("jitter") before back-shifting; this reduces high-frequency artifacts.

**Role in our framework.** DeepDream acts as a diagnostic for *Human Consistency* (H): if maximally activating synthetic images for a neuron are not recognized by humans as containing the intended concept, we discount that neuron's interpretability signal accordingly.

### B.4 LUCID FEATURE VISUALIZATIONS

Lucid Olah et al. (2017) produces more human-interpretable feature images via diversity objectives, TV/transform regularizers, and preconditioned gradients. We include one Lucid visualization per neuron to complement DeepDream: Lucid favors interpretability (often at lower absolute activation), while DeepDream probes whether maximal activation itself corresponds to human-recognizable content.

## C REPRODUCIBILITY

To ensure full reproducibility of our work, we provide open access to all code, data, and implementations used in this study. The complete codebase is organized across two GitHub repositories:

**Results, Metrics, and Visualizations:** All experimental code, metric implementations, and paper visualizations are available at: REDACTED FOR ANONYMITY

**Microscope Visualization Tool:** The interactive microscope visualization tool for exploring model interpretability is available at: `https://github.com/anonymous-bee?tab=repositories`

These repositories contain detailed documentation, installation instructions, and example usage to facilitate replication of our findings and enable further research in this area.

## D ETHICS

This framework is designed to improve AI safety through better interpretability assessment. More systematic evaluation of model components could help identify problematic behaviors and inform safer AI development.

We acknowledge that detailed knowledge of model internals could potentially inform adversarial attacks, but believe the benefits of interpretability research for AI safety outweigh these risks, particularly given the extensive existing literature on model internals.

