# OpenReview forum: "Beyond Anecdotal Evidence: A Systematic Framework for Evaluating Neuron Interpretability"
_ICLR.cc/2026/Conference — Submitted to ICLR 2026_

### Official Review · Reviewer_dfbX · 2025-10-29

**Soundness:** 2
**Presentation:** 2
**Contribution:** 2
**Rating:** 4
**Confidence:** 4

**Summary:**

This paper studies the problem of evaluating whether individual neuron captures specifc concept. To address this problem, the authors propose InterpScore, which summarizes Selectivity, Causality, Robustness and Human consistency.

**Strengths:**

1. This paper proposes new dimensions to evaluate whether neuron captures specific concept  beyond selectivity, including robustness, causality and human consistency.

**Weaknesses:**

1. The proposed metrics' ability to evaluate neuron-concept match is not well justified: For the causal impact metric, the only part related to concept is using concept image as test inputs. My intuition is this metric is more measuring the **importance of the neuron** to output embedding, instead of the quality of the concept. Similarly, the robustness metric is measuring robustness of neuron.
2. The empirical evidence is not fully convincing. The evidence 1 suggests InterpScore has better discrimination ability than that of selectivity alone. However, this is expected as more metrics are introduced and does not imply InterpScore is a better metric. Evidence 2 shows stability of InterpScore, which is a desired property but provides no information on whether it's good for evaluation neuron-concept pairs.

**Questions:**

1. Typos: Citation missing in Ln 66. Reference missing on Ln 388.
2. For causality and robustness evaluation, what happens if random images are used instead of images containing the specific concept? What about using images containing another concept? A full comparison is needed to justify those metrics capture information related to the concepts.
3. For human consistency, a potential problem with the evaluation protocol is it may favor general/ambiguious concept. For example, if a concept is "a photo", it may always be captured in highly activated images but this does not indicate it's a good concept.
4. A detailed case study will be helpful to justify the proposed InterpScore. For example, a case where selectivity only fails (high selectivity but not good concept) while the proposed InterpScore can identify it.

---

### Official Review · Reviewer_5gEr · 2025-10-29

**Soundness:** 2
**Presentation:** 3
**Contribution:** 1
**Rating:** 2
**Confidence:** 4

**Summary:**

This paper proposes a framework for aggregating four dimensions of unit-wise interpretability (selectivity, causal impact, robustness, human consistency) into a scalar interpretability score, which quantifies how interpretable an individual unit of a vision model is. The authors also implement a proof-of-concept for an openAI microscope-style website for network inspection.

**Strengths:**

- The paper is fairly well written and clear, apart from some open questions (see below).
- The paper demonstrates how the openAI microscope could be improved, in principle.
- Moving beyond just unit sensitivity is a good idea, in principle.

**Weaknesses:**

- As it stands, the paper does not really yield any new insights. The authors propose a way of computing interpretability scores, but don't motivate well why this would actually be useful. What do we do with these scores once we have them?
- The proposed approach is not easily scalable, because (a) the concepts used to evaluate neurons seem to be hand-picked and require manual labor and (b) the human consistency requires experiments that will be prohibitively expensive at scale.
- Averaging the four dimensions is a parameter-free, but still somewhat arbitrary choice. The scalar interpretability score could also be any other linear combination of the four dimensions.
- At the core of any neuron-level "interpretability score" should be how well humans can interpret the respective neuron. Operationalizing this notion is admittedly difficult, but the approach proposed in this work is not particularly convincing: 1. There could be fairly interpretable polysemantic neurons which activate for n concepts. In the current approach, the H-score of such units would not exceed 1/n, even if the concepts themselves were very clear. 2. Characterizing a unit by its behavior in the high-activity regime alone seems overly simplistic, since 95% of what the unit contributes to the network is by definition not captured by this operationalization.

This paper is generally well written, and attempts to move the field of interpretability towards quantifiable evaluations, which is laudable. The way in which this quantification is achieved is a reasonable starting point, and I appreciate the demo of the microscope-style website. But it is currently unclear how this approach could be scaled, and the paper does not sufficiently motivate the assignment of interpretability scores itself. It may be worth asking why the original microscope was discontinued: Even if the approach was scaled, what is the user story? How do we expect to derive new insights from such a microscope? Nothing in the paper is incorrect to the point of being outright wrong, but the contribution is not sufficient, in my opinion.

**Questions:**

- How are the concepts, for which images are then collected, chosen? This selection would have to be automated as well for the approach to scale.
- Not a question as such, but I suggest to rephrase the following sentence from the abstract, as it is hard to parse: "InterpScore reveals meaningful variation across neurons, about 14% dispersion where selectivity alone shows none, demonstrating that multi-axis evaluation surfaces distinctions overleaped by single metrics."
- There are missing references in line 65 and 388.
- In figure 3, the figure caption mentions points but no points are depicted.
- Very minor point and somewhat subjective, but I find it weird to have a "main research question"-block that does not contain a question.

---

### Official Review · Reviewer_Q2he · 2025-10-30

**Soundness:** 3
**Presentation:** 4
**Contribution:** 2
**Rating:** 4
**Confidence:** 2

**Summary:**

This paper proposes a 4-dimensional interpretation score (SCRH) to evaluate the interpretability of single neurons in neural networks.

The scores are illustrated on 10 neurons from a CLIP layer.
The paper demonstrates how going 4D provides more information than just reporting a neuron’s selectivity.

**Strengths:**

S1. The paper is clearly written, well organized, and conceptually well motivated. The narrative flows logically, and the motivation for studying interpretability is easy to follow.

S2. Each of the four proposed scores is well defined, intuitive, and supported by clear reasoning, making the framework easy to understand and apply.

S3. The running example of the sailing boat, and the well-designed figures, conveys the ideas very well.

S4. The authors reimplemented a previously closed-source tool, thereby contributing to open and reproducible research in interpretability. This is a really big strength.

**Weaknesses:**

W1. The experimental evaluation is limited. The analysis is conducted on only 10 neurons (are they cherry-picked for their high selectivity?), and drawn from a single model (clip). A broader evaluation, across multiple models, and on neurons that vary in selectivity, would strengthen the paper.

W2. The writing could be improved. Several references are missing or incomplete (e.g., “see:(?)” and “Table 1, App. ??”). This undermines the paper’s polish.

W3. The related work section seems insufficient. For instance, the paper does not describe psychophysics-inspired approaches to measuring interpretability (e.g., work by Zimmermann and others).

The paper does not describe approaches in language models, where the interpretability research is potentially more developed. What quantitative metrics exist there? The authors could discuss automated interpretability research for language models (eg, Bills et al., 2023; Cunningham et al., 2023; Gurnee et al., 2023; Bricken et al., 2023).

W4. Relatedly, the comparison to baselines is limited. Beyond the selectivity metric, it would be useful to discuss or empirically evaluate other established measures of interpretability, clarifying how the proposed scores compare in practice.

W5. Limitations section is missing.

**Questions:**

Q1. What are the concrete insights gained from the proposed 4D scoring framework? While it clearly captures more information than simple selectivity, it remains unclear what new understanding or _actionable_ conclusions this provides about neural representations. In other words: what does this added dimensionality tell us about interpretability?

Q2. How were the ten neurons used in the experiments selected? Were they chosen randomly or based on specific properties such as high selectivity or high variance? Clarifying the selection criteria is essential to assess the robustness and generality of the findings.

Q3. It would be helpful to include comprehensive tables in the related works listing existing interpretability metrics—one for language models and one for vision models. Such a summary would contextualize the contribution within the broader landscape of interpretability research and make it easier to compare approaches.

---

### Meta-Review · Area_Chair_M2DX · 2026-01-05

**Summary:**

This paper proposes a framework for evaluating the interpretability of neurons in vision models, with experiments on a small set of CLIP neurons. Reviewers generally agreed that the paper is clearly written and well motivated, but found the contribution limited. All reviews were borderline to negative, with consistent concerns about lack of insight, weak justification of the metrics, and limited empirical scope. No author rebuttal was provided. The AC has no basis for changing the reviewers' decisions.

**Reviewer Concerns:**

* evaluation is too small/narrow (10 neurons, one model; unclear selection)
* unclear what new insight or user value the score provides
* weak justification that the added axes actually measure neuron–concept alignment (vs generic importance/robustness)
* not scalable (manual concepts + human-consistency)
* aggregation into one score is arbitrary
* related work/baselines incomplete; some missing refs/placeholders

**Reviewer Scores:**

Reviewer scores are consistently negative or borderline and no rebuttal was submitted.

---

### Decision · Program_Chairs · 2026-01-26

Reject